# Shared decision-making interventions in the choice of antipsychotic prescription in people living with psychosis (SHAPE): Protocol for a realist review

Ita Fitzgerald[1,2]*, Laura J. Sahm[2,3], Jo Howe[4], Ian Maidment[4], Emma Wallace[5], Erin K. Crowley[2]

1 Pharmacy Department, St Patrick's Mental Health Services, Dublin, Ireland, 2 Pharmaceutical Care Research Group, School of Pharmacy, University College Cork, Cork, Ireland, 3 Pharmacy Department, Mercy University Hospital, Cork, Ireland, 4 School of Pharmacy, College of Health and Life Sciences, Aston University, Birmingham, United Kingdom, 5 Department of General Practice, School of Medicine, University College Cork, Cork, Ireland

* 118226904@umail.ucc.ie, itafitzgerald@rcsi.ie

**Data Availability Statement:** No results are reported - this is a protocol paper. Data availability statement not required.

## Abstract

### Background

Shared decision-making (SDM) has yet to be successfully adopted into routine use in psychiatric settings amongst people living with severe mental illnesses. Suboptimal rates of SDM are particularly prominent amongst patients with psychotic illnesses during antipsychotic treatment choices. Many interventions have been assessed for their efficacy in improving SDM within this context, although results have been variable and inconsistent.

### Aims

To generate an in-depth understanding of how, why, for whom, and to what extent interventions facilitating the application of SDM during antipsychotic treatment choices work and the impact of contextual factors on intervention effectiveness.

### Methods

This review will use realist review methodology to provide a causal understanding of how and why interventions work when implementing SDM during antipsychotic treatment choices. The cohort of interest will be those experiencing psychosis where ongoing treatment with an antipsychotic is clinically indicated. The review will take place over five stages; (1) Locating existing theories, (2) Searching for evidence, (3) Selecting articles, (4) Extracting and organising data and (5) Synthesizing evidence and drawing conclusions. An understanding of how and why interventions work will be achieved by developing realist programme theories on intervention effectiveness through iterative literature reviews and engaging with various stakeholder groups, including patient, clinician and carer representatives.

**Funding:** The author(s) received no specific funding for this work.

**Competing interests:** The authors declare no competing interests exist.

## Discussion

This is the first realist review aiming to identify generative mechanisms explaining how and why successful interventions aimed at improving SDM within the parameters outlined work and in which contexts desired outcomes are most likely to be achieved. Review findings will include suggestions for clinicians, policy and decision-makers about the most promising interventions to pursue and their ideal attributes.

## Introduction

Shared Decision-Making (SDM) is advocated as an ideal model of treatment decision-making in mental health and is a key component of person-centred care [1, 2]. SDM is a concept of non-paternalistic communication between patients and clinicians, and is most commonly defined as a process in which clinicians and patients work together to select treatments based on clinical evidence and the patient's informed preferences [2]. International mental health policies have increasingly advocated partnership models of mental health care, including the application of SDM in treatment decisions [1–3]. In the treatment of severe mental illnesses, the application of SDM may be particularly complicated [4]. Complexity in the application of the ideas and ideals of SDM may be particularly prominent in the treatment of schizophrenia and other enduring psychotic illnesses. During acute psychosis, a patient's decision-making capacity may be impaired to a significant degree, resulting in specific challenges for clinicians in the implementation of SDM in initial antipsychotic treatment decisions. Furthermore, the possibility of involuntary admission for treatment can create extreme forms of 'power asymmetry' and the importance of long-term antipsychotic adherence requires special attention to patient satisfaction with treatment [4].

The principles of SDM may, however, be particularly well-suited to the selection of antipsychotic drug treatment, an integral component of psychosis management [5]. Antipsychotic choice is considered largely a preference-sensitive decision [5, 6], where differences between antipsychotics primarily centre on differences in side effects rather than efficacy [7]. In such cases, choice of antipsychotic treatment is significantly influenced by the individual's preferences for likely side effects [6]. Such preference-sensitive decisions have been identified as an ideal target for SDM [6, 8]. Research has shown that the practice of SDM is highly acceptable amongst patients with enduring psychotic illnesses and psychiatrists [9–11], although differences in attitudes towards and subsequent participation in SDM have been identified in the case of the latter [12]. However, the practice of SDM has yet to be successfully adopted for routine use in psychiatric settings amongst patients with severe mental illnesses [13]. Suboptimal rates of SDM adoption are particularly prominent during antipsychotic treatment decisions amongst patients with psychotic illnesses [12, 14, 15]. Suggested reasons for low adoption rates of SDM in these contexts include clinicians' belief that patients with psychosis have low decisional capacity and cognitive (poor attention, deficits in working memory and verbal fluency) and motivational deficits [11, 16].

Studies assessing varying interventions aimed at improving the application of SDM in choice of antipsychotic treatment during psychosis have been undertaken [2]. Interventions have largely been modelled on the application of SDM models in somatic medicine, with additional design features to account for implementation within psychiatric settings [4]. Interventions assessed typically include a combination of decision aids [17, 18], educational interventions for patients and/or clinicians [14], and digital support tools [19–21]. To date, the

effect size of studied interventions has proven variable and inconsistent [2, 22] and positive results are generally smaller than in somatic medicine [4, 22, 23]. Reasons for varying results, including an understanding of which elements of efficacious interventions are hypothesized to be responsible for results and how they produced their effects, are largely missing from the literature. An understanding of these mechanisms is important to support increased and standardised application of SDM in antipsychotic treatment decisions.

As highlighted, applying SDM in choice of antipsychotic treatment during psychosis is associated with significantly more complexity than in somatic medicine [13]. Interventions are also expected to be embedded within existing complex environments and within systems which have traditionally used paternalistic, clinician-led decision making [1, 24]. Refinements and adaptations of traditional SDM models, in general and for local contexts, are likely needed to improve effectiveness, including consideration of contextual factors relating to patients, clinicians, and the clinical encounter [14]. Although information exchange is an essential element in facilitating SDM [24], the neglect of wider structural and contextual factors in the design and implementation of SDM in choice of antipsychotic treatment may be one reason for varying results and suboptimal implementation of SDM interventions [1, 15]. Thus, uncertainty exists about which intervention types to preferentially implement, characteristics of interventions to improve the likelihood of achieving desirable outcomes and the impact of different contexts on intervention effectiveness.

Any review that seeks to understand SDM interventions, including how they produce their effects, needs to look beyond the intervention and seek to make sense of the wider context. This need to account for context and to address questions of how and why interventions work provides the rationale for using realist review methods in this evidence synthesis [25]. Realist reviews aim to move from empirical observation to developing theoretical causal explanations to understand what it is about interventions that generate change (i.e., the mechanisms), and under what circumstances the mechanisms are triggered (i.e., the contexts), which result in changes in behaviour of the participants of the intervention (i.e., the outcome) [26]. These three elements i.e., context, mechanism, and outcome configurations (CMOC), are presented together as a programme theory which attempts to describe what needs to happen for the intervention to work. A realist approach to evidence synthesis offers distinctive strengths in addressing questions of what works, for whom, under what circumstances and how when attempting to develop complex interventions where generated outcomes are likely variable and context-dependent [27].

## Aims and objectives

This realist review aims to understand how interventions designed to improve SDM during antipsychotic treatment choices work and the impact of contextual factors on intervention success.

Review objectives include:

i. Review the literature to identify what interventions have been studied in improving SDM in antipsychotic treatment choices (e.g., choice of initial antipsychotic treatment, change of treatment, or continuation of initial treatment) amongst patients with a psychotic illness where SDM in the clinical context is preferred.

ii. Apply a realist logic of analysis to the literature to understand how and why interventions have or have not achieved their desired outcomes.

iii. Engage with key stakeholders including prescribers and clinicians/practitioners who support prescribing (pharmacists, nurses, social workers), patients, carers and family members to identify problems in engaging in SDM within the context outlined.

iv. Synthesize the findings into a realist programme theory outlining context-mechanism-outcome configurations to explain intervention effectiveness.

v. Provide recommendations on co-creating, tailoring, and implementing interventions to improve SDM during antipsychotic treatment choices in patients with a psychotic illness.

## Methods

Realist And Meta-narrative Evidence Synthesis: Evolving Standards (RAMESES) guidance will be followed throughout the review process [28]. While this review will be conducted and reported according to RAMESES standards for realist syntheses, the research team have also populated the PRISMA-P checklist to provide additional oversight in the methology of this review (see S1 Checklist). The review protocol has been registered with PROSPERO (CRD42023443783) on the 13/10/23 prior to review commencement. The review will follow five iterative stages based on Pawson's realist methodology, although the process of moving through the steps will proceed in a non-linear fashion [26, 29, 30] Fig 1 provides an overview of the five steps to be applied in this review [31, 32].

### Step 1: Locate existing theories

The rationale of this step is to identify a range of possible theories that explain how interventions aimed at improving SDM in decisions of antipsychotic treatment are supposed to work (and for whom), when they do work (or do not) and why they are not being used [25–27]. To locate these theories, we will (1) perform exploratory literature searches and (2) consult with members of the project team and stakeholder groups and draw on their experiential, professional and content knowledge. The project team represents multi- and inter-disciplinary professionals within psychiatry, academia and those with experience in education and clinical training. This step is more exploratory and aimed at quickly identifying the range of possible explanatory theories that may be relevant to the review question [25]. For this step, PubMed/ MEDLINE, PsycINFO and Open Grey will be used.

Following these searches and discussions with stakeholder groups, iterative discussions within the project team will be held to interpret and synthesize the different theories into an initial, coherent programme theory. Meetings of the project team and stakeholder groups will be held online via Microsoft Teams. Discussion may also be held via telephone calls and e-mail exchange. Detailed notes of all meetings will be kept to support programme theory development and refinement and to serve as a clear audit trail. From these processes, an initial programme theory for subsequent testing in the review will be developed.

**Stakeholder group—Clinicians/practitioners.** The clinician and practitioner stakeholder group will include representation from consultant psychiatrists, non-consultant psychiatric doctors, psychiatric nursing, psychiatric pharmacy,general practitioners and community pharmacy. We aim to identify 12–20 members will be identified through places of work, partnership organisations and through contacts of the project team. We will extend the membership as needed for testing of the emerging programme theory.

**Stakeholder group—Service users, informal and formal carer givers.** Those with lived experience of psychosis and taking antipsychotic medications will be identified within via the Service User Advisory Network in St Patrick's Mental Health Services (SPMHS), Dublin,

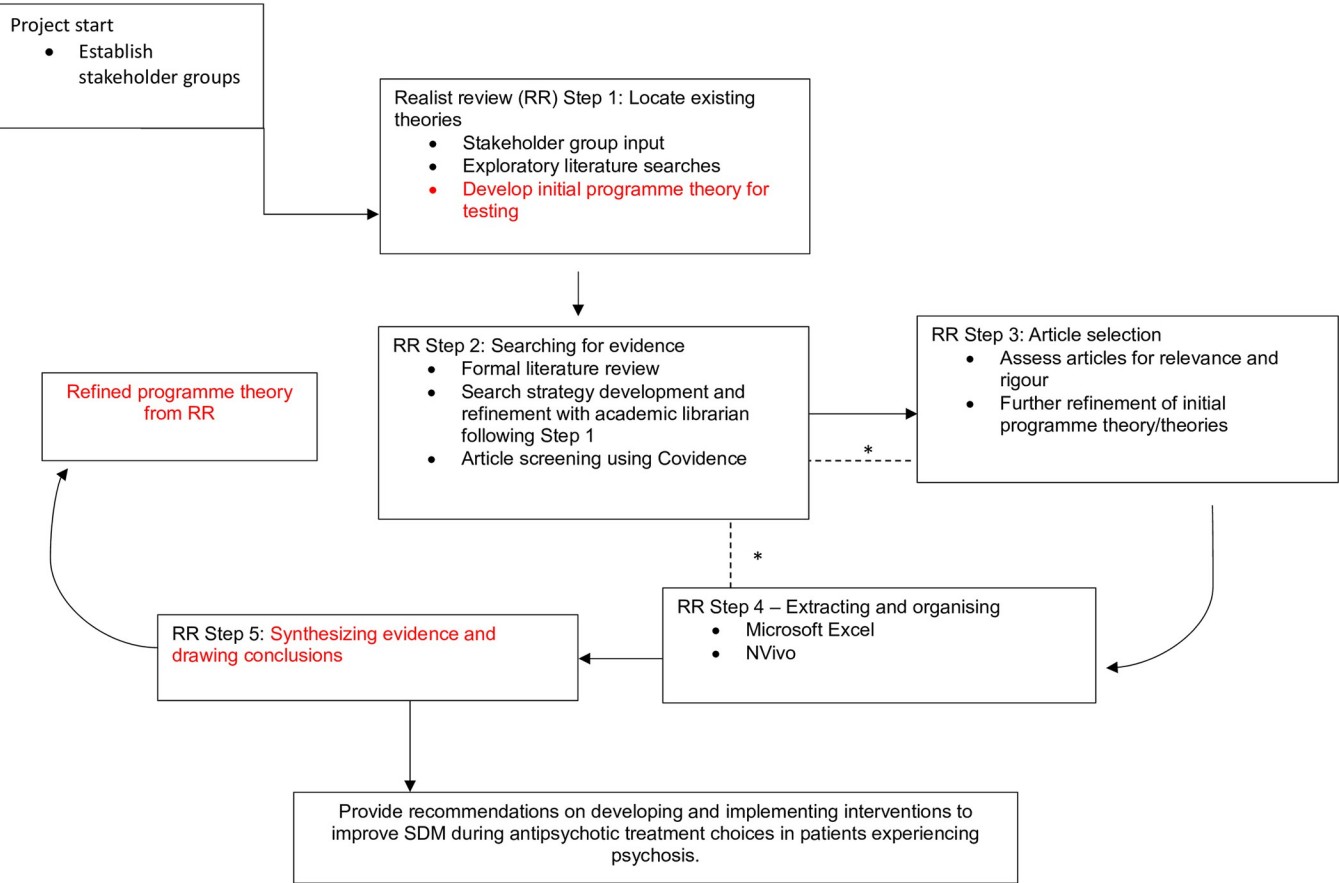

**Fig 1. Project flow diagram.** * Movement between steps if necessary to further refine programme theory. Adapted from review by Duddy et al. [31].

Ireland. Carers will be identified via the Family Members, Carers and Supporters Advisory Network in SPMHS. Both are established patient and carer stakeholder groups which afford local researchers the opportunity to set up advisory or consultation groups specific to the research project. We will also contact local charitable or public engagement organisations to recruit a diverse stakeholder group, if required. We aim to recruit 8–12 people across both cohorts. To ensure equal representation of the voice of both clinical expertise and expertise by experience, meetings of the clinician and lived experience stakeholder groups will meet separately. Similarities and differences in feedback from both groups will be shared with the other, and reasons for differences explored.

## Step 2—Searching for evidence

The purpose of this step is to find a relevant body of literature with which to further develop and refine the emerging programme theory formed in Step 1. Further programme theory refinement will use data identified via formal literature searches [25–27]. Once the initial programme theory has been developed as per Step 1, we will then be able develop the search strategy in full, as in other reviews [25, 26, 33, 34, 35]. For all searches, searching will be designed, piloted and conducted by one researcher (IF) with support from the project team and an academic librarian. We plan to conduct iterative searches of the literature with different search term concepts and permutations to capture the most relevant data relating to the emerging

**Table 1. Overview of inclusion and exclusion criteria.**

| | |
|---|---|
| Population | Include:<br>• Adult participants (aged 18–65) experiencing an episode of psychosis in the context of a psychotic illness where extended antipsychotic treatment is clinically indicated.<br>Exclude:<br>• Participants with treatment-resistant schizophrenia due to existence of clozapine as a preferred treatment choice amongst this cohort.<br>Participants experiencing substance/medication-induced psychosis or psychosis in the context of a general medical condition. |
| Intervention | Any intervention designed to facilitate SDM between clinicians and patients in decisions of antipsychotic treatment as part of psychosis management.<br>Given the role of collaborative goal setting and action planning in SDM in long-term conditions [36], alongside internalised stigma that can exist amongst those with mental illnesses [37], we will also include interventions that consist of SDM educational or training programmes for either patients and/or clinicians. |
| Comparator | Not applicable |
| Outcome | Outcomes of SDM processes have been assessed in a variety of different ways, relating to both process and outcome measurements [2]. This review will include outcomes relating to evidence of SDM application, including improved level of patient and clinician involvement in the decision-making process. Patient perceived involvement in decision-making can be assessed via many ways [2], for example the Shared-Decision Making Questionnaire (SDM-Q-9), the CollaboRATE scale or the Perceived Involvement in Care Scale (PICS) [38].<br>Other eligible outcomes relating to improving the likelihood of patient engagement in SDM specific to mental health settings [39], including patient-reported improved knowledge, empowerment, self-determination and satisfaction with treatment, will also be considered.<br>Other unanticipated outcomes may also be included in the review. For example, measures of patient satisfaction with care or quality of life measures may be relevant.<br>If SDM outcome assessment includes clinical outcomes, for example, improved adherence or reduced hospitalisation, these studies will also be included.<br>Outcomes also relating to physicians perceived involvement in SDM practices will be included, for example using the physician version of the SDM-Q-9-Doc [40]. |
| Timing | Use of interventions to inform choice of antipsychotic treatment (including initial treatment, change of treatment or continuation of treatment) as part of acute psychosis management. |
| Setting | Inpatient and outpatient settings, including community mental health teams and primary care settings to account for differing designs of mental health services internationally in the management of psychosis.<br>Forensic settings will be excluded.<br>The need for different programme theories for different settings will be considered by the research team. |

programme theory and any additional research that may add to the conceptual and contextual richness of the studies [27, 35]. Modification of the search strategy including terms searched, inclusion and exclusion criteria and databases used may be undertaken depending on the emerging programme theory. The proposed initial sampling frame to be used as the basis for the comprehensive literature search is outlined in Table 1.

Based on discussion with an academic librarian, we anticipate that we may need to search the following bibliographic databases: PubMed/MEDLINE, Embase, PsycINFO, CINAHL, the Cochrane library, Web of Science, Scopus and Sociological Abstracts. Additional searches for grey literature may be undertaken if required for programme theory refinement using the bibliographic databases Open Grey, ProQuest Dissertations, ResearchGate, Google Scholar, and Theses and DART-Europe-E-theses Portal. A combination of free-text and indexing search terms will be selected and adapted for the database being searched. There will be no restrictions on study type. Given the range of conceptual definitions of 'shared decision-making' and associated terminology, and the majority of research in this area being conducted amongst participants with schizophrenia [6, 18, 41], we will structure the search strategy according to schizophrenia and psychotic illness search terms and shared decision-making terms, with the

later derived from work conducted by Makoul et al. [42] Alerts for new articles which fit the search terms applied will be set to facilitate timely addition of new relevant articles during programme theory development. Only English language studies will be included due to study resources. A date restriction of 1990 to present will be applied. This reflects the timeline over which person-centred and recovery-focussed care in mental health became the dominant paradigms and associated application of SDM became advocated as the ideal model of treatment decision-making [1, 2].

Database searching will be supplemented by additional search methods. We will conduct backwards and forwards citation searching using Web of Science. We will also use 'cluster searching' techniques [33]. This includes 'sibling' (i.e. directly linked outputs from a single study) and 'kinship' (i.e. associated papers with a shared contextual or conceptual pedigree) papers [33, 43]. We will liaise with members of our stakeholder groups and additional links amongst the project team to recommend any potentially relevant documents. Searching will continue until sufficient data is found to conclude that the refined programme theory or theories are sufficiently coherent and plausible [27]. If the volume of the literature retrieved proves excessive, a variety of appropriate sampling strategies will be used (e.g. theoretical sampling, maximum variation sampling) to ensure that we have sufficiently focussed but relevant data for programme theory development [27, 33].

The results of all searches will be exported to Covidence systematic review software. Covidence is a web-based collaboration software platform that streamlines the production of systematic and other literature reviews (Covidence systematic review software, Veritas Health Innovation, Melbourne, Australia; see http://www.covidence.org). Following duplicate removal, screening of titles, abstracts and keywords of potentially relevant articles will be undertaken by one member of the research team (IF). A 10% random subsample of all studies will be reviewed independently by another researcher (LS/EC/IM/JH/EW) against the inclusion criteria for any systematic errors. Inclusion and exclusion criteria will be finalised by the project team following Step 1. Disagreements will be resolved by discussion and recourse to an independent member of the project team until consensus is achieved [26, 32].

## Step 3: Selecting articles

Screening of full-text articles identified for potential inclusion will be undertaken by one researcher (IF). Covidence software will also be used for this step. The selection of articles for final inclusion will primarily focus on relevance (whether data could contribute to some aspect of the testing and advancement of the programme theory) and rigour (whether the methods used to generate the relevant data are credible and trustworthy to warrant making changes to the programme theory) [27]. As in other studies and in line with best practice within realist review methodology [25–27, 32, 33], whilst we will consider the quality of each individual data source, we will conceptualise assessments of quality at a more global level. Judgements about quality will be made at the levels of data, argument, and programme theory. To operationalise such assessments, we plan to do the following [44]:

1. Find more than one source of data relevant to each aspect of the developing programme theory.

2. In cases of opinions etc; we will treat such data with scepticism initially until further supporting evidence is identified.

3. During building of programme theory, results will be presented initially partially, then in full, to stakeholder groups to aid in interpretation and to support development of a rigorous final programme theory.

4. We will seek to find to relevant substantive theory to support arguments made within the programme theory.

5. In cases of inadequate data, or use primarily of data judged to be of poor quality to support any aspect of the programme theory, we will cautiously report such statements with appropriate qualifiers regarding the quality of data.

6. We will provide access to supporting data taken from each listed data source at the point of publication to allow for critical appraisal by peers.

One researcher (IF) will read all included data sources that appear to contain data relevant to the realist analysis i.e., could inform some aspect of the programme theory. Reasons for exclusion of each study will be noted, for example if records are classified by the research team as having low relevance to the programme theory. For those articles deemed to meet the inclusion criteria, IF will retrieve the full text and classify documents into high and low relevance, depending on their relevance to the programme theory, and based on established methods previously employed [45, 46]. Briefly, full-text documents will be assigned a star rating of one to five, based on a global judgement of each document's relevance and rigour. Documents assigned a one- or two-star rating will be deemed irrelevant and not included in the review. Three-star documents will be deemed irrelevant for programme theory development but potentially useful for background material. Four-star documents will be classified as relevant for programme theory refinement. Five-star documents will be deemed the most conceptually rich or contextually thick and so, most relevant to CMOC development and programme theory refinement.

All 1–3-star documents will be reviewed again before finalisation of the programme theory (ies) to confirm appropriate classification [35, 43, 44]. A random subsample of the 10% of final documents for inclusion based on their assigned relevance and rigour judgement will be selected and assessed independently by another member of the research team (LS/EC/IM/JH/EW) to identify systematic errors [26, 32]. The remaining 90% of decisions will be made by IF, although a number of these may require further discussion/joint reading between the wider project team due to issues of uncertainty regarding relevance and/or rigour. Discussions will continue until consensus is reached.

## Step 4: Extracting and organising data

Data extraction and organisation will be undertaken by one researcher (IF) using Microsoft Excel. Study characteristics to be extracted include:

- Study details (publication year, location of study)

- Study objectives

- Intervention description

- Study design and quality markers (rigour, relevance)

- Study methods

- Sample characteristics (age, gender, ethnicity)

- Contextual factors (mechanisms) before the intervention was introduced

- Outcomes and how they were measured

The full texts of all included papers will be uploaded to NVivo qualitative data analysis software. Documents will be examined for data on how SDM interventions work by applying a

realist logic of analysis to relevant sections of the text. The synthesis of evidence will begin with conceptual coding. Sections of text will be coded in broad conceptual categories ('conceptual buckets') for example, developing therapeutic alliance, adequate information sources, beginning with the richest sources i.e. articles with the most potential to inform the programme theory. As the review progresses, these conceptual codes will be analysed to develop context-mechanism-outcome configurations (CMOC) [45]. Allocation of codes will be both inductive and deductive. Retroductive coding will also be applied i.e. where codes are created based on an interpretation of data to infer potential hidden causal mechanisms for outcomes [33, 45, 47]. Each new element of coded data will be used to refine the programme theory, as appropriate. As the theory is refined, included studies will be re-scrutinised for data relevant to the revised theory that may have been missed initially [33, 47]. This step will initially be completed by one researcher (IF) with support from other members of the team experienced in realist methodology (JH/IM). The project team will examine the viability of different CMOCs, experiment with varying formulations and work towards building the narrative of the evidence synthesis [32]. The developing programme theory will be confirmed with the rest of the project team iteratively and at defined stages. In the case of data extraction and coding of papers, a 10% random subsample of papers will be reviewed independently by other members of the research team (EW/LS/IM/JH/EC) as part of quality control measures. Any disagreements will be resolved via discussion until consensus is achieved.

## Step 5: Synthesising the evidence and drawing conclusions

To develop the final programme theory, we will move iteratively between the analysis of certain sections of included papers, stakeholder group interpretation and further iterative searching for data in the included studies to refine the programme theory and its subsections. The purpose of this step is to understand how mechanisms behave under the different contexts described within the review documents [27, 46]. We will move from data to theory to refine explanations about why certain interventions are effective (or not). This will include inferences about which mechanisms may be triggered in specific circumstances and contexts, as these are likely to be hidden and not explicitly or adequately referred to in the literature [28]. Relationships between context, mechanism and outcomes will be sought across articles included. In keeping with the application of a realist logic of analysis, a series of questions will be used to support the analysis and synthesis of data including [27, 33, 47]:

- Interpretation of meaning: if relevant and trustworthy, do the contents of the included document provide data that may be interpreted as functioning as context, mechanism or outcome?

- Interpretation and judgements about CMOCs: For example, what is the CMOC (partial or complete) for the data that has been interpreted as functioning as context, mechanism, or outcome? Are there further data to inform this particular CMOC contained within this source or other sources?

- Interpretations and judgements about programme theory: For example, how does this (full or partial) CMOC relate to the programme theory under development? Within the same document, are there data which informs how the CMOC relates to the programme theory?

When working through these questions, where appropriate, we will apply the following forms of reasoning to make sense of the data: juxtaposition of the data, reconciliation of the data, adjudication of the data and consolidation of the data [32, 33]. All members of the project team will be involved in generation of the final programme theory/theories.

### Ethics

Primary data will not be collected and therefore, ethical approval is not required for this review.

## Discussion

### Novelty of the review

This review will be the first realist review of the literature examining interventions aimed at improving SDM application during antipsychotic treatment choices amongst those with an enduring psychotic illness. The review will blend empirical research with the views, experience, and expertise of people with lived experience of psychosis, professionals and practitioners in this field, academics and topic experts. Systematic review findings suggest that several interventions are helpful in promoting the application of SDM within this context [2]. The literature has, however, focused on the effectiveness and impact of interventions, without considering underlying processes and contextual influences. There is a need for further evidence on how interventions work, for whom and under what circumstances to understand what can be done to maximise their chances of success. The review aims to identify those generative mechanisms underlying effective interventions and in which contexts are the desired outcomes most likely to be achieved. The findings of the review will enable us to provide suggestions for clinicians, policy and decision-makers about the most promising interventions to pursue and their ideal attributes, and what refinements are needed for local tailoring and implementation.

### Impact and dissemination

Review results will be used to inform future policy, research and practice in in this area. The research team will share findings through their networks and promote change beyond the end of the project. The findings of this realist review will also be made public through a peer-reviewed open access publication. Findings will be disseminated and shared through knowledge exchange with stakeholders and policymakers at a national and international level via conferences and personal communication. Key stakeholders within the project and wider team (including stakeholder groups) will be consulted to disseminate findings through their local and national networks. To increase the accessibility of the review findings, user-friendly summaries will be produced and tailored suitable for healthcare professionals, service users and their families. Use of social media platforms will be considered to increase engagement from the wider population.

## Supporting information

**S1 Checklist. PRISMA-P 2015 checklist.**
(DOCX)

## Acknowledgments

We would like to thank academic librarian Virginia Conrick for her assistance in the preparation of search strategy for this review.

## Author Contributions

**Conceptualization:** Ita Fitzgerald.

**Formal analysis:** Ita Fitzgerald.

**Investigation:** Ita Fitzgerald.

**Methodology:** Ita Fitzgerald, Laura J. Sahm, Jo Howe, Ian Maidment, Emma Wallace, Erin K. Crowley.

**Project administration:** Ita Fitzgerald, Laura J. Sahm, Erin K. Crowley.

**Resources:** Ita Fitzgerald, Laura J. Sahm, Jo Howe, Ian Maidment, Emma Wallace.

**Software:** Laura J. Sahm.

**Supervision:** Laura J. Sahm, Erin K. Crowley.

**Writing – original draft:** Ita Fitzgerald.

**Writing – review & editing:** Ita Fitzgerald, Laura J. Sahm, Jo Howe, Ian Maidment, Emma Wallace, Erin K. Crowley.

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
