## [Decision Letter · Decision Letter 0]

5 Apr 2024

PONE-D-23-34877Shared decision-making interventions in the choice of antipsychotic prescription in people living with psychosis (SHAPE): protocol for a realist reviewPLOS ONE

Dear Dr. Fitzgerald,

Thank you for submitting your manuscript to PLOS ONE. After careful consideration, we feel that it has merit but does not fully meet PLOS ONE’s publication criteria as it currently stands. Therefore, we invite you to submit a revised version of the manuscript that addresses the points raised during the review process.

We look forward to receiving your revised manuscript.

Kind regards,

Eleni Petkari

Academic Editor

PLOS ONE

2. We noticed you have some minor occurrence of overlapping text with the following previous publication, which needs to be addressed:

-https://doi.org/10.1371/journal.pone.0270028

In your revision ensure you cite all your sources (including your own works), and quote or rephrase any duplicated text outside the methods section. Further consideration is dependent on these concerns being addressed.

Reviewers' comments:

Reviewer's Responses to Questions

**Comments to the Author**

1. Does the manuscript provide a valid rationale for the proposed study, with clearly identified and justified research questions?

Reviewer #1: Yes

Reviewer #2: Yes

2. Is the protocol technically sound and planned in a manner that will lead to a meaningful outcome and allow testing the stated hypotheses?

Reviewer #1: Yes

Reviewer #2: Yes

3. Is the methodology feasible and described in sufficient detail to allow the work to be replicable?

Reviewer #1: Yes

Reviewer #2: Yes

4. Have the authors described where all data underlying the findings will be made available when the study is complete?

Reviewer #1: Yes

Reviewer #2: No

5. Is the manuscript presented in an intelligible fashion and written in standard English?

Reviewer #1: Yes

Reviewer #2: Yes

6. Review Comments to the Author

You may also provide optional suggestions and comments to authors that they might find helpful in planning their study.

Reviewer #1: Dear Authors,

I have reviewed your manuscript "Shared decision-making interventions in the choice of antipsychotic prescription in people living with psychosis (SHAPE): protocol for a realist review." It is my pleasure to recommend its acceptance.

Your manuscript stands out for its adherence to the RAMESES project principles, contributing to its methodological strength. The choice of a realist review is particularly apt given the noted heterogeneity in outcomes of traditional review methods in this field. Additionally, your inclusive approach of involving various stakeholders, notably patients, is praiseworthy and enriches the study's depth and applicability.

Your work addresses a crucial aspect of mental health research and has the potential to significantly impact clinical practices and inform policy-making in the treatment of psychosis. I commend your efforts in enhancing patient-centered care through this research.

Congratulations on a well-executed study. I look forward to its publication and the impact it will have on the field.

Best regards,

Antonio Di Francesco

Reviewer #2: Thank you for the opportunity to review your work.

The authors present a protocol for a realist review study that aims to generate an in-depth understanding of interventions to facilitate the application of shared decision-making during the choice of antipsychotic drug treatment. The authors conclude that their review will be the first realist review that will identify generative mechanisms that explain how and why successful interventions aimed at improving shared decision-making in antipsychotic drug treatments and will identify which contexts desired outcomes of shared decision-making interventions would be most likely to be achieved.

I commend the authors on the work put forth to complete this protocol. This is an important topic that highlights an ongoing clinical issue of identifying the best approach to facilitating shared decision-making for antipsychotic treatment in psychotic illness. The protocol manuscript is overall clear, well-written, and reports an appropriate and rigorous methodologic approach. I provide the following suggestions for strengthening the manuscript.

1. The review protocol appears to be focused on shared decision-making on antipsychotic drug treatments in the setting of psychotic illness. However, this is not clear within the aims and/or methods of the abstract. Would the authors be able to clarify this detail in their manuscript?

2. The inclusion of multiple stakeholder groups including clinicians and service users as well as informal and formal caregivers is a strength of the study protocol and I commend the authors for using this comprehensive approach. When engaging with stakeholder groups that include service users and informal and formal caregivers, how will the authors ensure that perspectives of these stakeholder groups are prioritized and integrated among the larger stakeholder group of clinicians and practitioners?

3. The authors identify the patient population of interest to include adult participants aged 18-65 years. Would the authors be able to clarify the rationale for exclusion of those participants over the age of 65 years?

4. The authors report that during full-text screening that articles will be classified into high and low relevance based on previous established methods. It would be helpful to the reader to briefly explain how articles will be defined as having high or low relevance in the study.

5. How do the authors intend to evaluate rigor (i.e., trustworthiness) when completing qualitative data analysis during evidence synthesis?

I thank the authors for the opportunity to review this manuscript.

7. PLOS authors have the option to publish the peer review history of their article (what does this mean?). If published, this will include your full peer review and any attached files.

Reviewer #1: **Yes: **Antonio Di Francesco

Reviewer #2: No

---

## [Author Response · Author response to Decision Letter 0]

3 May 2024

Dear Dr Petkari,

Many thanks to you and your reviewers for providing positive feedback regarding our manuscript and your constructive comments. We have responded to reviewer comments below in red. Sections of the manuscript which we have subsequently amended are contained here and have been highlighted within this response for ease of identification of amendments.

Kind regards,

Ita

Reviewer #2: Thank you for the opportunity to review your work.

1. The review protocol appears to be focused on shared decision-making on antipsychotic drug treatments in the setting of psychotic illness. However, this is not clear within the aims and/or methods of the abstract. Would the authors be able to clarify this detail in their manuscript? Thank you. We have modified the methods section of our abstract to reflect the patient population our review findings will address. The methods section in the abstract (manuscript line 13-14) now reads as follows:

Methods: This review will use realist review methodology to provide a causal understanding of how and why interventions work when implementing SDM during antipsychotic treatment choices. The cohort of interest will be those experiencing psychosis where ongoing treatment with an antipsychotic is clinically indicated.

2. The inclusion of multiple stakeholder groups including clinicians and service users as well as informal and formal caregivers is a strength of the study protocol and I commend the authors for using this comprehensive approach. When engaging with stakeholder groups that include service users and informal and formal caregivers, how will the authors ensure that perspectives of these stakeholder groups are prioritized and integrated among the larger stakeholder group of clinicians and practitioners? Thank you for raising this, it is an important point that requires due consideration. Meetings of both groups will be held separately. Clinicians will meet separately to the group representing those with lived experience of psychosis and informal and formal carers. We recognised that inviting those with clinical expertise and expertise by experience to meet together could very reasonably reduce openness amongst both groups when considering factors that hinder and facilitate shared decision making, particularly during acute psychiatric care. However, the same topics will be discussed within the respective groups. Similarities and differences in interpretation of review results, next steps in programme theory development or other tasks assigned to each stakeholder group will subsequently be discussed with the other. A peer support worker representing both lived experience of psychosis and experience of working within acute psychiatric care contexts is also a member of the clinician stakeholder group to give prominence to the voice of lived experience in clinician stakeholder group meetings.

To address your important point further within the manuscript, we have modified lines 171-174. This section now reads as follows:

To ensure independent representation of the voice of both clinical expertise and expertise by experience, meetings of the clinician and lived experience stakeholder groups will meet separately. Similarities and differences in feedback from both groups will be shared with the other, and reasons for differences explored.

3. The authors identify the patient population of interest to include adult participants aged 18-65 years. Would the authors be able to clarify the rationale for exclusion of those participants over the age of 65 years? Yes, thank you. The participant population of interest to this review are: 

• Inclusion:

o Adult participants (aged 18-65) experiencing an episode of psychosis in the context of a psychotic illness where extended antipsychotic treatment is indicated.

• Exclusion:

o Participants with treatment-resistant schizophrenia due to existence of clozapine as a preferred treatment choice amongst this cohort.

o Participants experiencing substance/medication-induced psychosis or psychosis in the context of a general medical condition.

In our introduction section (lines 52-54), we outline ongoing suboptimal rates of shared decision-making amongst those with a severe mental illness in routine psychiatric care. Our rationale not to include adults aged >65 years relates to the average age at which a diagnosis of a severe mental illness is provided (usually between 20-30 years of age for schizophrenia, bipolar affective disorder, and major depressive disorder) to patients and thus, where antipsychotic treatment will be both started and continued. In such cases, the absence of shared decision-making will be most impactful. Other causes of psychotic symptoms more likely in those >65 years of age e.g., delirium, dementia, mild cognitive impairment are all situations where ongoing antipsychotic treatment is very unlikely to be clinically appropriate. 

Additionally, whilst the absence of shared decision-making is applicable to those >65 years of age, ‘mechanisms’ which are responsible for the success of interventions and their interaction with contextual factors are likely to be different amongst younger versus older adults. For example, if a complex intervention offers patients additional information on antipsychotic treatment choices as one intervention strategy to improve the application of SDM, if increased confidence amongst patients aged <65 to engage in discussions is identified as the likely causal mechanism, it is reasonable to assume that this context-sensitive mechanism may not be equally applicable amongst someone who is 25 versus 75. Therefore, in line with the purpose of our review and intended recommendations for practice regarding implementation of SDM in SMI management, we will exclude studies solely addressing SMI in those >65 years of age. We hope this adequately addresses your query.

4. The authors report that during full-text screening that articles will be classified into high and low relevance based on previous established methods. It would be helpful to the reader to briefly explain how articles will be defined as having high or low relevance in the study. Thank you, this is useful feedback. We agree and have adapted lines 312-322 of our manuscript accordingly. This now reads as follows:

For those articles deemed to meet the inclusion criteria, IF will retrieve the full text and classify documents into high and low relevance, depending on their relevance to the programme theory, and based on established methods previously employed.[45] Briefly, full-text documents will be assigned a star rating of one to five, based on a global judgement of each document’s relevance and rigour. Documents assigned a one- or two-star rating will be deemed irrelevant and not included in the review. Three-star documents will be deemed irrelevant for programme theory development but potentially useful for background material. Four-star documents will be classified as relevant for programme theory refinement. Five-star documents will be deemed the most conceptually rich or contextually thick and so, most relevant to CMOC development and programme theory refinement. All 1–3-star documents will be reviewed again before finalisation of the programme theory (ies) to confirm appropriate classification.[32,43,44] A random subsample of the 10% of final documents for inclusion based on their assigned relevance and rigour judgement will be selected and assessed independently by another member of the research team (LS/EC/IM/JH/EW) to identify systematic errors.[26, 33]

5. How do the authors intend to evaluate rigor (i.e., trustworthiness) when completing qualitative data analysis during evidence synthesis? Many thanks for raising this. We believe from working with methodologists on this review there is no short answer to your question. Data are relevant in a realist review when they can help develop, corroborate, refute, or refine aspects of realist programme theory (or theories). The norm within realist reviews is that a programme theory that meets suggested quality criteria of simplicity, consilience, and analogy is underpinned by multiple arguments based on analysis and interpretation of many data sources, not all of which will represent empirical studies. We anticipate within this review the role of policy and regulation will be important for prescriber perceived willingness and ability to engage in shared decision-making – a key facilitator to making shared decision-making a reality within routine psychiatric settings. Thus, policy documents and texts addressing the impact of policy on service delivery and prescriber behaviour will be of high relevance. Whilst the option exists to use critical appraisal tools in the case of relevant qualitative and quantitative studies, no such tool will exist in the case of many documents relevant to the development of a coherent programme theory, for example, policy documents. 

Thus, we will consider the quality of each individual data source, but, as outlined in the RAMESES guidelines, we will conceptualise assessments of quality at a more global level. Judgements about quality will be made at the levels of data, argument, and theory. To operationalise such assessment, we plan to do the following (as outlined in Wong, 2018 – see ref 44):

1. Find more than one source of data relevant to each aspect of the developing programme theory.

2. In cases of opinions etc; we will treat such data with scepticism initially until further supporting evidence is identified. 

3. During building of programme theory, results will be presented initially partially, then in full, to stakeholder groups to aid in interpretation and to support development of a rigorous final programme theory.

4. We will seek to find to relevant substantive theory to support arguments made within the programme theory. 

5. In cases of inadequate data or use primarily of data judged to be of poor quality to support any aspect of the programme theory, we will cautiously report such statements with appropriate qualifiers regarding the quality of data. 

6. We will provide access to supporting data taken from each listed data source at the point of publication to allow for critical appraisal by peers. 

We have also modified our manuscript now to address your important feedback. Lines 266-284 now read as follows:

As in other studies and in line with best practice within realist review methodology,[25-27, 33,34] whilst we will consider the quality of each individual data source, we will conceptualise assessments of quality at a more global level. Judgements about quality will be made at the levels of data, argument, and programme theory. To operationalise such assessments, we plan to do the following:[44]

1. Find more than one source of data relevant to each aspect of the developing programme theory.

2. In cases of opinions etc; we will treat such data with scepticism initially until further supporting evidence is identified. 

3. During building of programme theory, results will be presented initially partially, then in full, to stakeholder groups to aid in interpretation and to support development of a rigorous final programme theory.

4. We will seek to find to relevant substantive theory to support arguments made within the programme theory. 

5. In cases of inadequate data or use primarily of data judged to be of poor quality to support any aspect of the programme theory, we will cautiously report such statements with appropriate qualifiers regarding the quality of data. 

6. We will provide access to supporting data taken from each listed data source at the point of publication to allow for critical appraisal by peers. 

We have also now included the following book chapter by Wong within our reference sources (ref. 44): Data Gathering in Realist Reviews Looking for needles in haystacks | Semantic Scholar

---

## [Editor Report · Decision Letter 1]

15 May 2024

Shared decision-making interventions in the choice of antipsychotic prescription in people living with psychosis (SHAPE): protocol for a realist review

PONE-D-23-34877R1

Dear Dr. Fitzgerald,

We’re pleased to inform you that your manuscript has been judged scientifically suitable for publication and will be formally accepted for publication once it meets all outstanding technical requirements.

Kind regards,

Eleni Petkari

Academic Editor

PLOS ONE
---

## [Editor Report · Acceptance letter]

17 May 2024

PONE-D-23-34877R1 

PLOS ONE

Dear Dr. Fitzgerald, 

I'm pleased to inform you that your manuscript has been deemed suitable for publication in PLOS ONE. Congratulations! Your manuscript is now being handed over to our production team.

Kind regards, 

on behalf of

Dr. Eleni Petkari 

Academic Editor

PLOS ONE